# Cold-Active Starch-Degrading Enzymes from a Cold and Alkaline Greenland Environment: Role of Ca^2+^ Ions and Conformational Dynamics in Psychrophilicity

**DOI:** 10.3390/biom15030415

**Published:** 2025-03-14

**Authors:** Malthe Kjær Bendtsen, Jan Stanislaw Nowak, Pedro Paiva, Marcos López Hernández, Pedro Ferreira, Jan Skov Pedersen, Nicolai Sundgaard Bekker, Elia Viezzi, Francesco Bisiak, Ditlev E. Brodersen, Lars Haastrup Pedersen, Athanasios Zervas, Pedro A. Fernandes, Maria Joao Ramos, Peter Stougaard, Mariane Schmidt Thøgersen, Daniel E. Otzen

**Affiliations:** 1Interdisciplinary Nanoscience Center, Gustav Wieds Vej 14, 8000 Aarhus, Denmark; malthe@inano.au.dk (M.K.B.); jsn@mbg.au.dk (J.S.N.); marcoslh@inano.au.dk (M.L.H.); jsp@chem.au.dk (J.S.P.); viezzielia@gmail.com (E.V.); 2LAQV, REQUIMTE, Departamento de Química e Bioquímica, Faculdade de Ciências Universidade do Porto, Rua do Campo Alegre, s/n, 4169-007 Porto, Portugal; pedro.paiva@fc.up.pt (P.P.); peferreira@fc.up.pt (P.F.); pafernan@fc.up.pt (P.A.F.); mjramos@fc.up.pt (M.J.R.); 3Department of Chemistry, Aarhus University, Gustav Wieds Vej 14, 8000 Aarhus, Denmark; 4Department of Chemistry and Bioscience, Aalborg University, Fredrik Bajers Vej 7H, 9220 Aalborg, Denmark; nicolaibekker@gmail.com (N.S.B.); lhp@bio.aau.dk (L.H.P.); 5Department of Molecular Biology and Genetics, Aarhus University, Universitetsbyen 81, 8000 Aarhus, Denmark; francesco.bisiak@gmail.com (F.B.); deb@mbg.au.dk (D.E.B.); 6Department of Environmental Science, Aarhus University, 4000 Roskilde, Denmark; az@envs.au.dk (A.Z.); pst@envs.au.dk (P.S.); math@zealand.dk (M.S.T.)

**Keywords:** psychrophilic enzymes, protein stability, molecular dynamics

## Abstract

Cold-active enzymes hold promise for energy-efficient processes. Amylases are widely used in household and industrial applications, but only a few are cold-active. Here we describe three novel secreted amylases, Rho13, Ika2 and I3C6, all from bacteria growing in the cold and alkaline ikaite columns in Greenland. They all hydrolyzed starch to smaller malto-oligomers, but only Rho13 and Ika2 hydrolyzed cyclodextrins, and only Ika2 displayed transglycosylation activity. Ika2 forms a stable dimer, while both Rho13 and I3C6 are mainly monomeric. They all have optimal active temperatures around 30–35 °C and significant enzymatic activity below 20 °C, but Rho13 and I3C6 had an alkaline optimal pH, while Ika2 was markedly acidophilic. They showed complex dependence on Ca^2+^ concentration, with the activity of Rho13 and I3C6 following a bell-shaped curve and Ika2 being unaffected; however, removal of Ca^2+^ reduced the stability of all three enzymes. Loss of structure occurred well above the temperature of optimal activity, showing the characteristic psychrophilic divorce between activity and stability. MD simulations showed that Ika2 did not have a well-defined Ca^2+^ binding site, while Rho13 and I3C6 both maintained one stably bound Ca^2+^ ion. We identified psychrophilic features as higher levels of backbone fluctuations compared to mesophilic counterparts, based on a lower number of internal hydrogen bonds and salt bridges. This increased fluctuation was also found in regions outside the active site and may provide easier substrate access and accommodation, as well as faster barrier transitions. Our work sheds further light on the many ways in which psychrophilic enzymes adapt to increased catalysis at lower temperatures.

## 1. Introduction

Exploring cold environments for new cold-active enzymes can lead to the discovery of novel enzyme variants with unique properties and functionalities. These enzymes may exhibit distinct substrate specificities, thermal stabilities, or catalytic efficiencies, offering opportunities for tailored enzyme engineering and customization to specific industrial applications. The ability of cold-active enzymes to function effectively at low temperatures provides flexibility in process design. Industrial processes can be carried out at ambient temperatures or under refrigeration, eliminating the need for energy-intensive heating systems. This minimizes the risk of thermal degradation of sensitive substrates or products and reduces operational costs. Novel energy-efficient processes are required in industry as well as in private households to contribute to a more sustainable society. One contribution to this reduction in emissions is to apply truly cold-active enzymes to detergents to make washing at low temperatures even more efficient.

Cold-active enzymes for detergents are already industrially applied today. However, most of these enzymes display optimal activity at temperatures between 20 and 45 °C [1], with marginal activity at 5–20 °C. This is usually compensated for by over-dosing the amount of enzyme used. To move beyond these simplistic solutions, we need to discover novel, truly cold-active enzymes from natural environments that provide in-depth knowledge of cold activity [2]. This knowledge can then be mobilized to either exploit these new enzymes directly or modify currently used industrial enzymes to increase their activity at low temperatures and to reduce the amount of enzyme required.

α-amylases (EC 3.2.1.1) are a major class of enzymes used in laundry detergents, as well as in the production of textiles, biofuel, feed, food, and beverages. They catalyze hydrolysis of α-1,4 glycosidic bonds in starch and starch-related polysaccharides, including amylopectin, amylose, cyclodextrins, glycogen, and pullulan [3,4,5]. Amylases are also relevant for bioremediation. Cold-active amylases are particularly interesting because of their ability to degrade organic matter, for example in waste treatment in cold and temperate regions [6]. Some cold-active amylases even possess higher specific activity than their meso- or thermophilic counterparts [6].

α-amylases are structurally grouped in four different glycoside hydrolase (GH) families: GH13, GH57, GH119, and GH126 mentioned in the CAZy database of carbohydrate acting enzymes [3]. The largest family is GH13, whose members form a (β/α)_8_ barrel structure and a conserved catalytic triad consisting of two Asp and one Glu, or in a few cases one Asp, one Glu, and one His [7]. In addition α-amylases, the GH13 family also contains endo- as well as exo-acting amylase-family enzymes with a wide range of different substrate specificities, including pullulanases (catalyzing hydrolysis of (1→6)-α-D-glucosidic linkages in pullulan, amylopectin and glycogen), neopullulanases (α-amylases specifically catalyzing the hydrolysis of α-1,4-glucosidic bonds in pullulan), and cyclomaltodextrinases (catalyzing the linearization of cyclomaltodextrin into maltodextrin) [8,9,10,11,12,13].

An attractive feature of cold-active amylases is that they are active at moderate as well as low temperatures. Most known amylases are either meso- or thermophilic, and only a limited number of cold-active amylases are known. These include an α-amylase from the soil bacterium *Bacillus subtilis* [14], an α-amylase from the Himalayan soil bacterium *Bacillus cereus* [15], a salt-tolerant α-amylase from the marine bacterium *Zunongwangia profunda* [16], and the well-characterized α-amylase from the Antarctic marine bacterium *Alteromonas haloplanktis* A23, which shows optimal amylolytic activity at 30 °C and maintains 18% activity at 3 °C [17,18,19,20].

Since most amylases are exported, they will be exposed to conditions which may differ from the cytosolic environment in terms of ionic strength or pH. Amylases with significant residual activity at low temperature have so far been found in aquatic environments, and all show relatively broad pH profiles, with optimal pH values around 10 (*Bacillus subtilis* N8 and *Bacillus cereus* GA6) and 7 (*Zunongwangia profunda* and *Alteromonas haloplanktis* A23). Alkaliphilic profiles are particularly useful for detergent enzymes, since detergent-based washing usually takes place at around pH 9–10 [21].

To discover novel cold-active, alkali-stable amylases, we investigated the only environment on Earth known to be both cold and alkaline, namely the ikaite columns in the Ikka Fjord, SW Greenland. These tufa columns, consisting of the meta-stable mineral ikaite (a hydrated form of calcium carbonate), are formed due to precipitation of calcium carbonate from highly alkaline spring water (pH 10.4) mixing with the cold seawater in the Ikka Fjord [22]. They are inhabited by diverse microbial communities [23]. Most of the isolated bacterial strains and recombinantly expressed enzyme genes obtained from environmental DNA from the columns show significant extracellular enzymatic activity at high pH levels [24,25,26].

Here we identify and provide a detailed characterization of two novel starch-degrading enzymes from these ikaite columns, namely Ika2 and Rho13. In parallel, we extend the characterization of I3C6, another amylase isolated from the ikaite columns [27], to allow direct comparison of their enzymatic and biophysical properties, with a particular focus on their psychrophilicity. I3C6 was identified in a metagenomic analysis rather than from functional screening of a single bacterial isolate [27] and had a reported pH optimum around 7–9 and a temperature optimum around 10–15 °C, losing activity within 20 min at 45 °C while retaining > 70% of activity at as low as 1 °C, and with a pH optimum at pH 8–9. I3C6 activity was stimulated by Ca^2+^ but inhibited by Zn^2+^ and Cu^2+^. We analyzed product formation by pulsed amperometric detection and observed endo-active release of maltose (G2) and larger oligosaccharides from amylopectin, amylose, and hydrolyzed starch, as well as hydrolysis of smaller maltooligosaccharides. The three enzymes also vary in their quaternary organization, pH profiles, and dependence on Ca^2+^ ions, but they all show loss of activity at temperatures well below the loss of structure. This characteristic psychrophilic trait may be linked to their higher levels of backbone fluctuations compared with mesophilic homologues. We propose that this provides easier substrate access and binding, as well as facilitating efficient conformational transitions across local minima during catalysis.

## 2. Materials and Methods

### 2.1. Identification of Samples

All samples originated from the ikaite columns in the Ikka Fjord but were sampled at different times between 2006 and 2019, and the enzymes were detected and isolated by different approaches, as described below.

*Amylase Ika2*: Samples of ikaite columns were collected by divers in the Ikka Fjord in the summer of 2019, with permission from the Government of Greenland (License no. G19-032), and stored at −20 °C. To isolate bacteria from the ikaite columns, samples of 0.2–0.5 g of ikaite were drilled by hand from the interior of the column samples. The drilled samples were suspended in 1/10 strength R2 broth [per liter: 0.05 g yeast extract, 0.05 g Bacto peptone, 0.05 g Bacto casamino acids, 0.03 g sodium pyruvate, 0.03 g KH_2_PO_4_, 0.005 g MgSO_4_ * 7 H_2_O, 1 g NaCl] adjusted to pH 10 with 50 mM Na_2_CO_3_/NaHCO_3_, and aliquots were used for culturing experiments. Bacterial strains from the sampled ikaite material displaying amylase activity were picked from agar plates and passaged to 1/10 strength R2 agar [1/10 strength R2 broth supplemented with 15 g/L agar, no glucose] adjusted to pH 10 and supplemented with 0.5 g/L AZCL-amylose (Megazyme, Wicklow, Ireland). The plates were incubated at 5 °C for 3 months, and colonies producing a halo of solubilized chromogenic amylose were re-streaked on the same substrate as above until strain purity was achieved.

*Amylase RhoGH13_36* (denoted below): The gene sequence was identified by screening the genome sequence of the ikaite bacterium *Rhodonellum psychrophilum* [28,29] (NCBI accession number AWXR00000000) for putative amylase-encoding genes (see details below).

*Amylase Amy_I3C6_* (denoted below): This enzyme was previously identified and described by Vester et al. [27] (acc. no. KJ790257).

### 2.2. Identification of Amylase-Coding Genes and Homology Alignment

To identify amylase-coding genes in the cultured isolate (Ika2), the strain was subjected to whole-genome sequencing. DNA was prepared for sequencing using a Nextera XT DNA Library Preparation Kit (Illumina, Inc., San Diego, CA, USA) and sequenced on an Illumina MiSeq. The assembled reads, along with the published genome of *R. psychrophilum,* were annotated using PROKKA [30] and analyzed using Geneious Prime (v. 2022.2.2) [31]. To identify potential amylolytic enzyme–encoding genes in the genomes of Ika2 and Rho13, a local sequence list containing the amino acid sequences of characterized amylases representing all relevant GH groups (GH13, 14, 15, 57, 97, 119, and 126) in the CAZy database was created and blasted against the genome sequence (tblastn) with a max E-value of 10 to also include low-scoring sequence hits. All potential hits were analyzed for sequence-based and in silico structural similarities to known enzymes and enzyme-encoding genes using the Blast module (blastn, blastp, and tblastn) of Geneious Prime against the nucleotide collection (nr/nt) database, Protein Data Bank (PDB), Swiss-Prot, Uni-Prot [32,33], and HHPred (MPI Bioinformatics Tools) [34].

The selected “best candidates” were analyzed in InterPro [35] to confirm their classification, and signal peptides were further confirmed using SignalP 6.0 [36].

### 2.3. Cloning, Expression, and Purification

*Ika2*: A gene predicted to encode an enzyme with sequence similarities to the α-amylases neopullulanase and cyclomaltodextrinase was selected for expression. The gene was amplified using gene-specific primers with overhangs designed for In-Fusion^®^ cloning (Takara Bio Inc., Kusatsu, Japan) into plasmid pET28a(+), including a C-terminal 6x His-tag: Ika-amy2-F (5′-AGGAGATATACCATGTTATTAGAAGCAATTTACCATCGGC-3′) and Ika-amy2-R (5′-GTGGTGGTGGTGGTGGGCACCGCTTTTGATGATGAC-3′). The primers used to amplify the vector were pET28a-F (5′-CACCACCACCACCACCACT-3′) and pET28a-R (5′-CATGGTATATCTCCTTCTTAAAGTT-3′) (vector-specific overhangs are underlined).

The DNA fragments were amplified using 2X Phusion High-Fidelity PCR Master Mix with HF Buffer (Thermo Scientific) and the following PCR protocol: 1 cycle at 98 °C for 30 s, 25 cycles at (98 °C for 10 s, 55 °C for 1 min, 72 °C for 30 s/kb), and 1 cycle at 72 °C for 5 min (amylase gene)/10 min (vector). The PCR products were visualized on a 1% agarose gel and purified using a Monarch^®^ DNA Gel Extraction Kit (New England Biolabs, Ipswich, MA, USA). In-Fusion^®^ cloning was carried out according to the protocol supplied by Takara Bio using 5X In-Fusion Snap Assembly Master Mix (Takara Bio Inc.). In brief, 2 µL 5X Master Mix was mixed with 50–100 ng purified vector DNA and 50–100 ng purified insert DNA and incubated at 50 °C for 15 min, followed by incubation on ice for 15 min. The reaction directly transformed into *E. coli* BL21(DE3). The transformed cells were grown on LB agar with 50 µg/mL kanamycin (Km50) at 37 °C overnight.

*Rho13 and I3C6*: Genes encoding the two putative amylases were codon-optimized for *E. coli* and directly cloned into plasmid pET28a(+) including a C-terminal 6x His-tag by GenScript (Rijswijk, The Netherlands). Signal sequences were excluded. The plasmids were transformed into *E. coli* BL21(DE3) for expression. A single colony from each of the three clones was inoculated into 10 mL LB Km50 and incubated overnight at 37 °C with 200 rpm shaking. Each overnight culture was used to inoculate 1 L of LB Km50, which was incubated at 37 °C with 200 rpm shaking for 4 h. The cultures were subsequently incubated at RT with 200 rpm shaking for 20 min before adding 1 mM isopropyl ß-d-1-thiogalactopyranoside for induction. The induced cultures were incubated at 18 °C with 200 rpm shaking overnight. Cells from the induced overnight cultures were harvested by centrifugation (10 min, 4500 rpm, 4 °C). The supernatant was discarded, and the pellet was resuspended in 12.5 mL Buffer A (50 mM Tris, 50 mM NaCl, pH 8). The cells were lysed by sonication (5 min, 20% amplitude, pulses of 20 s with 20 s on ice in between). The lysates were centrifuged for 30 min at 15,000× *g* (4 °C), and the supernatant was filtered through a 0.45 µm Millipore filter. The filtrate was added to a 15 mL purification column consisting of 4 mL Ni-NTA beads (Thermo Fisher Scientific, Waltham, MA, USA) previously charged with 4 mL 1M NiSO_4_ and equilibrated with washing buffer (buffer A + 20 mM imidazole). After addition of the filtrate, the column was incubated for 15 min and washed with 15 mL wash buffer, and the amylase was eluted with 7 mL elution buffer (Buffer A + 400 mM imidazole).

The eluates were desalted on PD-10 columns (Cytiva, Marlborough, MA, USA) before further purification using an Äkta Pure Protein Purification System (Cytiva) on a Superdex-200 Increase column with a flow rate of 0.5 mL/min. Purifications were carried out in either (1) 100 mM HEPES with 150 mM NaCl and 2 mM CaCl_2_ at pH 6.5 or (2) 50 mM MOPS with 50 mM NaCl and 2 mM CaCl_2_ at pH 7.5. Fractions giving rise to absorption at 280 nm were examined for activity using an MBTH assay (see below) and run on SDS-PAGE gels to assess their purity with Mark12™ Unstained Standard (Thermo Fisher Scientific) (Ika2) or PageRuler™ Unstained Protein Ladder (Thermo Fisher Scientific) (Rho13 and I3C6) as standards. The purified enzymes were further concentrated using an Amicon ultra 30 K spin filter. Mw, pI, and extinction coefficients were calculated using the ProtParam tool (https://web.expasy.org/protparam/ (accessed on 24 February 2023)). Protein concentrations were measured using a Nanodrop ND-1000 (Thermo Fisher Scientific, Waltham, MA, USA).

### 2.4. Determination of Carbohydrate Breakdown Products

As amylase substrates we used potato starch, beta- and gamma-cyclodextrins (β-CD and γ-CD) (Wacher), maltose (reagent grade, VWR, Søborg, Denmark), maltooligosaccharide preparations of dp3 (maltotriose > 90%), dp6 (maltohexaose > 65%), and dp7 (maltoheptaose > 60%) (Merck, Søborg, Denmark). For hydrolysis reactions, each enzyme was incubated with 5 g/L starch or CD in 50 mM MOPS buffer pH 7.5 and 2 mM CaCl_2_, along with either Ika2 (135 nM) and I3C6 (300 nM) at 30 °C or Rho (135 nM) at 35 °C. Total volume: 1 mL, sampling 90 mL at 0, 2, and 4 h. Due to the low activity of I3C6, it was also sampled at 24 and 48 h. For potential transglycosylation assays, I3C6 was incubated with 11.6 mM glucose/fucose in combination with maltoheptaose and Rho13 with 11.6 mM glucose/fucose in combination with maltohexaose. Ika2 was incubated in 11.6 mM glucose only, 11.6 mM glucose in combination with maltose, maltotriose, maltoheptaose, and γ-CD, and 11.6 mM fucose in combination with glucose and γ-CD. All reactions were stopped by adding 0.5 M NaOH to the sample 1:1 (*v*/*v*). For all assays, controls with only enzyme and no substrate or substrate without enzyme were included in the experiments. All samples were analyzed by High Performance Anion Exchange Chromatography with Pulsed Amperometric Detection on an ICS-6000 system (Dionex™, Thermo Fisher Scientific, Waltham, MA, USA). The system was fitted with a CarboPac PA1 column at 25 °C, flow rate 0.4 mL/min, and injection volume 25 µL. The eluents were (A) 1 M sodium hydroxide, (B) 1 M sodium acetate, and (C) milliQ-water. The eluent contained constantly 10% (A) and from 0–5 min 1% (B) and 89% C); from 5–30 min a linear gradient of (B) increasing to 30% at the expense of (C), from 30–35 min 90% (B) at the expense of (C), and from 35–45 min original conditions, equilibrating the system for the next analysis. We used a detector gold working electrode pH-Ag-AgCl reference and a Dionex standard waveform potential cycle for carbohydrates. Standards included glucose, maltose, maltooligosaccharides of dp3-7, β-CD, and γ-CD.

### 2.5. Activity Assays

*MBTH amylase activity assay:* Enzymatic activity was determined by measuring the amount of reducing carbohydrate ends in a 3-methyl-2-benzothiazolinone hydrazone hydrochloride (MBTH) assay [37]. For each experiment, 10 mg/mL soluble potato starch (Sigma Aldrich, St. Louis, MO, USA) was dissolved in an assay-specific buffer (see below), followed by acid-catalyzed hydrolysis (adjustment of pH to 1.8 using 6 M HCl followed by 45 min at 95 °C). The pH was then adjusted to the desired assay pH with 6 M NaOH, and the concentration was reduced to 5 mg/mL. The activity assays were performed at 21 °C in a reaction mix consisting of 135–200 nM amylase and 4.5 mg/mL hydrolyzed starch in the appropriate assay buffer. Every 30 s, 25 µL of the reaction mix was removed, and the reaction was stopped with 25 µL 0.5 M NaOH. Subsequently, 25 µL MBTH reagent (1.5 mg/mL MBTH and 0.5 mg/mL DTT) was added, and the mixture was incubated at 80 °C for 15 min. Immediately after removal from the heater, 50 µL developing agent was added (0.5% (FeNH_4_(SO_4_)_2_)·12H_2_O, 0.5% sulfamic acid, 0.25 M HCl, or 0.5 M HCl for the pH assay), and the samples were left to develop and cool for at least 15 min. After cooling, 20 µL sample was diluted in 180 µL Milli-Q H_2_O in a 96 well plate, and absorbance was measured at 620 nm. If the absorbance was measured to be higher than the standard curve, the samples were further diluted 20-fold in Milli-Q H_2_O. Absorbance was converted to reducing ends using a standard curve made from a dilution series of 0.122–62.5 nmol maltose. Activity/initial velocity was determined from linear regression of the increasing number of reducing ends over 330 s.

*pH optimum:* The assay buffer was 100 mM bis-tris propane, 50 mM NaCl, and 2 mM CaCl_2_. The starch solution was prepared in assay buffer, and the pH was adjusted to 6–9.5 (6, 6.5, 7, 7.5, 8, 8.5, 9, 9.5) with NaOH and HCl. The enzyme was mixed with assay buffer of the relevant pH and allowed to incubate for 15 min at RT before starch was added. Initial velocity was measured over 5.5 min. Enzyme concentrations: Rho13: 200 nM, I3C6: 135 nM, Ika2: 136 nM.

*NaCl optimum:* Assay buffer: 50 mM MOPS, 2 mM CaCl_2_, pH 7.5, NaCl: 0–1000 mM (0, 10, 20, 50, 100, 250, 500, 1000 mM). The substrate and assay buffer were prepared without NaCl, but the pH was adjusted with NaOH (given MOPS’ pKa of 7.2, the buffer contributes ca. 34 mM to ionic strength). The desired NaCl concentration was achieved using a 2 M NaCl stock in assay buffer. Initial velocity was measured over 5.5 min. Enzyme concentrations: Rho13: 200 nM, I3C6: 140 nM, Ika2: 136 nM.

*Ca^2+^ optimum*: Assay buffer: 50 mM MOPS, 50 mM NaCl, pH 7.5, CaCl_2_: 0–20 mM (0, 0.01, 0.05, 0.2, 0.5, 2, 5, 20 mM). The substrate and assay buffer were prepared without CaCl_2_, and the desired CaCl_2_ concentration was achieved using a 100 mM CaCl_2_ stock in assay buffer. Enzymes were buffer-exchanged by size exclusion chromatography (SEC) purification on an Äkta pure system on a Superdex-200 Increase column in assay buffer without CaCl_2_. Enzyme concentrations: Rho13: 200 nM, I3C6: 135 nM, Ika2: 136 nM. The activity as a function of Ca(II) concentration showed a bell-shaped curve and was fitted to Equation (1) [38]:(1)$$V_{max}^{apo}=\frac{\frac{V_{max}^{apo}\cdot K_{L_{1}}}{\left[Ca\left(II\right)\right]}+V_{max}^{L_{1}}+\frac{V_{max}^{L_{2}}\cdot [Ca(II)]}{K_{L_{2}}}}{\frac{K_{L_{1}}}{\left[Ca\left(II\right)\right]}+1+\frac{\left[Ca\left(II\right)\right]}{K_{L_{2}}}}$$

Here, $V_{max}^{apo}$, $V_{max}^{L_{1}}$, and $V_{max}^{L_{2}}$ are the maximum velocities of the enzyme with zero, one, and two ligand molecules bound, respectively. $K_{L_{1}}$ and $K_{L_{2}}$ are dissociation constants for binding of the first and second ligand molecule, respectively. This model is simplified, as the proteins might have several binding sites.

*Temperature optimum:* Assay buffer: The enzyme was mixed with buffer and placed on ice until assay start. The enzyme and the substrate were individually heated to assay temperature (5, 10, 15, 20, 25, 30, 35, 40, 45, 55 °C) for 15 min before the substrate was added to the enzyme solution, and the temperature was kept co81nstant during the experiment. Every 30 s, 25 µL of the reaction was stopped in 0.5 M NaOH, and the activity was measured by MBTH assay. Activity was determined as a linear fit of the initial 5.5 min of the reaction. The enzyme was prepared in 50 mM MOPS buffer pH 7.5 without CaCl_2_ but was then diluted 1:4 in assay buffer containing CaCl_2_ for 15 min before assay start. Enzyme concentrations: Rho13: 115 nM, I3C6: 50 nM, Ika2: 580 nM.

### 2.6. Biophysical Analysis

*Circular Dichroism (CiD):* The secondary structure of each enzyme (200 µL, 0.2 mg/mL) in 10 mM MOPS, 50 mM NaCl, 2 mM CaCl_2_ with or without 3 mM EDTA was measured on a Chirascan-plus qCD spectrometer (Applied Photophysics, Surrey, UK). The sample was exposed to a temperature ramp of 1 °C/min from 8–90 °C while recording spectra at 200–260 nm in 1 nm steps. The bandwidth was set to 1 nm and the measurement time to 0.5 s/nm. The recorded temperature was measured directly in the cuvette. Wavelength spectra are shown with buffer background subtracted. Thermal scans were fitted in OriginPro 2024 following a two-transition model (Equation (2)) to determine the global melting temperature *t*_m_ (indicated in °C).(2)$$CD_{app}\left(t\right)=\frac{\left(bs_{N}+as_{N}*t\right)*e^{∆H_{1}*\frac{1-\frac{t+273.15}{t_{m1}+273.15}}{R*\left(t+273.15\right)}}+\left(bs_{I}+as_{I}*t\right)+\left(bs_{D}+as_{D}*t\right)*e^{-∆H_{2}*\frac{1-\frac{t+273.15}{t_{m2}+273.15}}{R*\left(t+273.15\right)}}}{e^{∆H_{1}*\frac{1-\frac{t+273.15}{t_{m1}+273.15}}{R*\left(t+273.15\right)}}+1+e^{-∆H_{2}*\frac{1-\frac{t+273.15}{t_{m2}+273.15}}{R*\left(t+273.15\right)}}}$$

With the CiD_app_ signal as a function of temperature and *t*_m1_ and *t*_m2_ representing the two melting points and Δ*H*_1_ and Δ*H*_2_ representing the melting enthalpies respectively, R is the gas constant, and *bs_N_ + as_N_*t*, *bs_I_ + as_I_*t*, and *bs_D_ + as_D_*t* the CiD signal for the native, intermediate, and denatured species, respectively.

*Differential scanning fluorimetry:* Loss of tertiary structure during heating was monitored with differential scanning fluorimetry on a NanoTemper Prometheus Panta with 7 µL enzyme (0.1 mg/mL) in 50 mM MOPS, 50 mM NaCl, 2 mM CaCl_2_. Excitation was at 280 nm, and fluorescence emission at 350 nm and 330 nm was measured in parallel at a scan rate of 1 °C/min from 15 to 95 °C. Reversibility of unfolding was measured in the same run by decreasing the temperature to 15 °C at the same scan rate. The 350 nm/330 nm fluorescence ratio was plotted against temperature and fitted to a two-state unfolding equation (Equation (3)) in GraphPad Prism 10.0.2.(3)Iapp=IF+IFB∗t+IU+IUB∗t∗e−∆Hunf∗1−ttmRt1+e−∆Hunf∗1−ttmRt

Here, *t* is the temperature, *t*_m_ is the melting temperature, Δ*H* is the unfolding enthalpy, and *R* is the gas constant. Fluorescence intensity as a function of temperature is described by *I*_F_ + *I*_FB_**t* and *I*_U_ + *I*_UB_**t*, for the folded and unfolded species, respectively.

*Size exclusion chromatography (SEC):* Chromatograms from the SEC purifications were plotted and compared to the elution profiles of three standard globular proteins (lysozyme (14.31 kDa), carbonic anhydrase (28.98 kDa and 57.96 kDa), and BSA (66.46 and 132.93 kDa)). The elution volume with the highest absorbance was defined as the elution volume for each peak. The data were plotted as log_10_ of the MW as a function of elution volume. Estimated MW was calculated from the linear regression on the plot of five peaks from the standard proteins. The MW of the α-amylases was calculated from their sequence, excluding the signal peptide from Rho13 but including the C-terminal His-tag extension LEHHHHHH (Rho13 and I3C6) or HHHHHH (Ika2).

*Structure prediction:* AlphaFold2 was used to predict the structure of I3C6 and a dimer structure of Ika2 using the multimer setting used for SAXS [39]. The top result was relaxed with AMBER. Rho13 monomer and dimer structures were predicted using CoLabFold [40,41] The predicted models had high local and global confidence prediction scores.

*Small-angle X-ray scattering (SAXS) data collection:* All SAXS data were obtained at Aarhus University using an in-house flux-optimized Bruker (Karlsruhe, Germany) SAXS NanoStar instrument with a Ga liquid metal jet source from Excillum (Kista, Sweden) and home-built scatterless slits [42]. All measurements were performed for 1800 s at 25 °C. Buffer and noise contributions were measured and subtracted. The resulting data were normalized to absolute scale using scattering from water and logarithmically rebinned. Scattering intensity is described as a function of the modulus of the scattering vector *q*, *q* = (4π sin(θ))/λ, where 2θ is the scattering angle, and λ = 1.34 Å is the radiation wavelength for the Ga radiation. SAXS data were acquired using different concentrations for each protein to check for possible concentration effects. All SAXS data treatment and analysis procedures described in the text were carried out using the SUPERSAXS software package (J.S. Pedersen and C.L.P. Oliveira, unpublished). Model-independent analysis was performed using an indirect Fourier transformation routine [43,44] to acquire the pair distance distribution function, *ρ*(*r*), from which the maximum distance (*D_max_*), forward scattering (*I*(0)), and radius of gyration (*R*_g_) were obtained. The value *I*(0) was used to estimate the molecular weight, *M*, of the scatterer particle using the equation $M=\frac{I\left(0\right)}{c∆\rho_{m}^{2}}$, where *c* is the mass concentration per milliliter and Δρ_m_ is the excess scattering length density per mass, which can be approximated as $2.0\times 10^{10}\frac{cm}{g}$ for proteins. From this calculated value and the theoretical molecular weight of each monomer, an oligomerization state can be determined.

*SAXS data analysis:* Model-dependent analysis was performed using structures predicted by AlphaFold in combination with the in-house programs *wlsq_symxv6XL* [45] and *wlsq_lin*. The program *wlsq_symxv6XL* calculates theoretical scattering curves of PDB structures with contribution from a hydration layer by applying the Debye equation using a Gaussian form factor for all non-H atoms. Since the data are normalized to absolute scale, the concentration can be used as a scale parameter and therefore compared with the value obtained from absorbance measurements. Additionally, possible oligomerization states can be optimized from the starting structure using different symmetries, which is more thoroughly described in [46]. This feature was applied to Rho using P2 symmetry, which corresponds to a simple two-fold symmetry. The program *wlsq_lin* is used to fit a linear combination of different oligomer scattering profiles in cases where SAXS model–independent analysis and SEC suggested the coexistence of multiple species.

### 2.7. Molecular Dynamics Simulations

To assess the overall stability and flexibility of I3C6, Ika2, and Rho13, we conducted Molecular Dynamics (MD) simulations of these enzymes and some close homologs. Prior to the simulations, the predicted structures of the psychrophilic enzymes were aligned with the structures of their mesophilic/thermophilic counterparts. I3C6 (monomer) was aligned with the α-amylase from *Bacillus stearothermophilus* (PDB: 1HVX [47]; backbone RMSd ≈ 1.03 Å), Ika2 (dimer) was aligned with the neopullulanase from *Geobacillus stearothermophilus* (PDB: 1J0H [48]; backbone RMSd ≈ 2.14 Å), and Rho13 (monomer) was aligned with the α-amylase from *Halothermothrix orenii* (PDB: 1WZA [49]; backbone RMSd ≈ 2.65 Å). After careful analysis of the aligned structures, we copied the coordinates of the Ca^2+^/Na^+^ ions from the homologs’ crystallographic structures to the respective psychrophilic structures (Figure 1). The protonation state of all residues at the optimum pH for each enzyme (see Results) was initially based on the results of the p*K*_a_ predictor software PROPKA 3.5.0 [50]. Additionally, the chosen protonation states were carefully analyzed and validated through visual inspection of the residues and their surrounding environment.

The ANTECHAMBER module of AMBER 18 [51] was employed to neutralize (with Na^+^ counterions) each amylase and to place it in the center of a rectangular box of TIP3P water molecules [52], whose faces were located at least 15 Å from the surface of each enzyme. All simulations were conducted using GROMACS 2021.5 software [53] and the AMBER ff14SB force field [54]. Prior to the MD simulation runs, each system was submitted to a multi-step minimization protocol using the steepest descent algorithm [55]. In the first step, both the enzyme and the modeled Ca^2+^/Na^+^ ions were restrained, allowing minimization of all water molecules; in the second minimization step, the side chains of all residues were relaxed; in the third and last step, the entire system, except the modeled Ca^2+^/Na^+^ ions, was minimized. After minimization, the obtained systems were used as the starting points of MD simulations, in which the LINCS algorithm [56] was applied to constrain all bonds involving hydrogen atoms. This made it possible to apply an integration time step of 2 fs. In all simulations, the Verlet cut-off scheme was used in combination with a non-bonded cut-off value of 10 Å. Furthermore, periodic boundary conditions were considered, and the Coulomb interactions were treated using the Particle Mesh–Ewald scheme [57]. The minimized systems were first heated for 100 ps to the target temperature (30–35 °C, according to the optimal temperature of each psychrophilic enzyme, see Section 3 Results) using the V-rescale thermostat [58] in the *NVT* ensemble. During this heating phase, all protein atoms and all modeled Ca^2+^/Na^+^ ions were restrained by a harmonic potential with a force constant of 1000 kJ/mol/nm^2^. Subsequently, a 2 ns equilibration phase was performed in the *NPT* ensemble to relax the system’s density to 1.0 bar, which was accomplished using the Berendsen barostat [59]. The enzymes’ backbones and the modeled Ca^2+^/Na^+^ ions were restrained throughout the entire equilibration phase. Finally, a 400 ns production phase was performed for each system at constant temperature and pressure (*NPT* ensemble) using the V-rescale thermostat and a Parrinello–Rahman barostat [60]. During this stage, no positional restraints were applied to the system. This three-phase MD protocol was carried out in three replicates, for a total production time of 1.2 µs per amylase system.

To analyze the structural motions of the enzymes, the MD trajectories of the production phase obtained for the three replicates were concatenated to a continuous trajectory of 1.2 μs using the *gmx trjcat* tool. For each system, the root-mean-square deviation (RMSD) of the backbone atoms relative to the initial minimized structure was calculated with the *gmx rms* tool to evaluate the enzyme’s stability. The conformational flexibility of the enzymes was evaluated with per-residue root-mean-square-fluctuation (RMSF) analysis, performed on the backbone atoms using the *gmx rmsf* tool. The resulting values provide a comparative measure of psychrophilic versus mesophilic enzyme local and global flexibility. To minimize the strong dependence of the calculated RMSF values on the average structure used as a reference, this particular metric was calculated individually for each replicate and subsequently combined to an average ± standard deviation value. The compactness of the enzyme structures was assessed by calculating the *R_g_* using the *gmx gyrate* tool. Typically, a higher *R_g_* is suggestive of a less compact and more flexible structure [61]. The solvent-accessible surface area (SASA), calculated with the *gmx sasa* tool, served as another descriptor of structural compactness, with higher SASA values indicating increasing solvent exposure, which strongly influences residue flexibility [62]. Principal component analysis (PCA) was employed to reduce the dimensionality of the molecular motions observed in the simulations. For this, the covariance matrices of the atomic fluctuations of the backbone atoms were constructed and diagonalized using the *gmx covar* tool. Subsequent projection of the trajectory onto the first two eigenvectors was performed using the *gmx anaeig* tool. These eigenvectors represented roughly 50% of the total conformational freedom, as evidenced by the cumulative contribution of all eigenvectors. Visualization and image generation of the structural conformations and dynamic trajectories were carried out using VMD 1.9.4 [63] and PyMOL 2.3.0 [64].

## 3. Results & Discussion

### 3.1. Isolation, Identification, and Recombinant Expression of Amylases

Here we report on the enzymatic and biophysical properties of two new psychrophilic amylases, Rho13 and Ika2, and compare them to our previously reported psychrophilic amylase I3C6 [27]. The predicted structures of all three enzymes are shown in Figure 1. We start by describing how we identified the two enzymes.

#### 3.1.1. Rho13

A putative GH13-family amylase-coding gene (referred to as Rho13) was identified in the published, PROKKA-annotated genome sequence of the cold-active bacterium *R. psychrophilum* [28,29] identified in ikaite column samples. The gene consisted of 1545 bp, including a 5′-region encoding a 16 amino acid predicted signal sequence. A BlastP search against PDB showed highest similarity to CAZY GH13 subfamily 36, with the highest similarity to an α-amylase from *Halothermothrix orenii* (pdb: 1WZA) (42.80%) and an amylase from *Eubacterium rectale* (pdb: 7JJN) (33.53%). The fragment encoding the mature sequence excluding the signal sequence (1497 bp, i.e., 499 residues) was cloned into a pET28 vector encoding a C-terminal His-tag and recombinantly expressed in *E. coli* BL21(DE3). This allowed us to purify Rho13 on Ni-NTA columns followed by size exclusion chromatography (SEC), leading to single-band purity (Figure 2a).

#### 3.1.2. Ika2

Bacteria-containing material was extracted from ikaite columns in the Ikka Fjord, Greenland, and incubated for 3 months at 5 °C on diluted R2-based substrate containing AZCL-amylose. An amylase-positive bacterial strain was isolated in this way. To identify the gene encoding the active amylase, the bacterial isolate was subjected to full genome sequencing [31], which allowed us to assign the strain to a novel taxon within the *Bacillaceae* related to *Anaerobacillus*, *Paenibacillus*, and *Alkalihalobacillus*. The PROKKA-annotated genome was mined for putative amylase-encoding genes, and an open reading frame consisting of 1737 bp was identified as a gene potentially encoding a cyclomaltodextrinase, α-amylase, α-glucosidase, or neopullulanase. In a BlastP search against PDB, the translated gene sequence shared highest similarity with four starch-degrading enzymes belonging to the GH13 subfamily 20 (GH13_20), namely a cyclomaltodextrinase (which linearizes cyclodextrins) from *Bacillus* sp. (pdb: 1EA9) (56.61%) [13], a maltogenic amylase (which releases maltose from starch) from *Thermus* sp. (pdb: 1GVI) (55.2%) [65], a neopullulanase (which hydrolyzes α-1,4 linkages in pullulan through transglycosylation) from *Geobacillus stearothermophilus* (pdb: 1J0H, 1J0J) (53.9%) [48], and α-amylase II (which cleaves starch into smaller oligosaccharides) from *Thermoactinomyces vulgaris* (pdb: 1WZM) (46.5%) [66,67,68]. The sequence similarity was rather low, but starch-degrading enzymes are known to be a highly diverse group [69], and several of them exhibit multiple catalytic activities. For example, maltogenic amylases hydrolyze both α-d-(1,4)- and α-d-(1,6)-glycosidic bonds and exhibit transglycosylation activation on a range of mono- and disaccharide acceptors [65,70]. In silico folding of the putative protein sequence using HHpred showed that it was indeed able to fold into an amylase 3D structure. To verify the amylolytic activity of the identified putative GH13 enzyme (denoted Ika2), the predicted gene was recombinantly expressed with a C-terminal His-tag and purified from *E. coli* in a manner similar to Rho13 (Figure 2b).

### 3.2. Structural and Biophysical Analysis of the Amylases

In the following text, we report on the enzymatic activity of the two amylases as a function of temperature, pH, CaCl_2_, and NaCl, as well as presenting a structural and biophysical analysis in vitro and in silico. We compare them to the psychrophilic amylase I3C6, which was isolated from a metagenomic analysis of ikaite columns and was expressed and purified recombinantly (Figure 2c). This allowed a direct and simple comparison between the psychrophilic properties of the three enzymes. Detailed mechanistic studies providing *K*_M_ and *k*_cat_ values will be reported elsewhere in a separate study.

#### 3.2.1. Substrate Specificity and Cleavage Products

We first investigated which carbohydrate substrates were processed and what products were formed by the three enzymes, guided by the activities of the enzymes they had been found to be related to (see previous section). To measure hydrolytic activity, the amylases were incubated with starch or cyclodextrins (β-CD and γ-CD), the products were analyzed by pulsed amperometric detection (PAD), and peaks were assigned as described [71]. In addition, we checked for potential transglycosylation activity by incubating the enzymes with either glucose or fucose (a deoxy sugar in the L conformation) in the presence of a maltooligosaccharide preparation containing maltoheptaose (dp7, where dp is degree of polymerization) or maltohexaose (dp6). For a better overview, representative profiles are shown in Figure 3a–c, while additional PAD elution profiles are shown in Appendix A.

##### Ika2

After 4 h of incubation, Ika2 hydrolyzed starch to glucose, maltose, and a series of maltodextrins with increasing dp (Figure 3a). In the same period, β-CD was hydrolyzed to glucose, maltose, and maltotriose, as well as smaller amounts of maltodextrins up to dp7 and a product tentatively assigned to isomaltose [72,73] (Appendix A). A similar pattern was observed with γ-CD (Figure 3b), which was degraded to glucose, maltose, and maltodextrins up to dp8; maltooctaose (corresponding to linearized γ-CD) was degraded further after 4 h incubation. Incubation with fucose and the maltooligosaccharide preparation containing maltoheptaose showed that fucose remained unconverted, while glucose and maltose were formed at the expense of maltooligomers (maltohexaose and -heptaose in particular) (Figure 3c). In addition, products appeared tentatively assigned to isomaltose. Products eluting with retention times between maltose and dp5 are presumably positional isomers of dp3–dp5 and remained after 48 h of incubation, suggesting transglycosylation activity (see arrows in Figure 3c). The degradation of cyclodextrins points to endo activity, and the degradation of starch and malto-oligomers resulted in the formation of glucose as one of the products.

##### Rho13

Starch was hydrolyzed, and formation of glucose, maltose, and maltotriose was observed, with further degradation of the latter resulting in glucose and maltose as the main products after 4 h of incubation. Unlike Ika2 (and I3C6, see below), maltodextrins with dp> 6 seemed absent. β-CD was degraded to glucose, maltose, and maltotriose, as well as small amounts of maltotetraose (Appendix A). γ-CD was degraded to glucose, maltose, and maltooligosaccharides of dp3–7 after 4 h of incubation (Appendix A), as well as positional isomers of maltose (isomaltose), maltotriose, and maltotetraose, though the distribution of dp3–4 isomers differed subtly from that of Ika2. Overall, γ-CD seemed a better substrate than β-CD, given the larger concentrations of products. Incubation with glucose and a maltooligosaccharide preparation primarily containing dp4–6 showed only a small increase in glucose concentration but a significant production of maltose with degradation of maltooligomers already after 4 h, with maltotriose being produced after 4 h but degraded after 48 h (Appendix A). Rho13’s degradation of cyclodextrins also indicates endo activity, like Ika2, though the ability to degrade maltotriose and maltotetraose illustrates a degree of exo activity as well.

##### I3C6

No activity of the enzyme was observed on cyclodextrins compared with the zero-time sample (Appendix A). There was a lower level of starch degradation than with Ika2, consistent with the low general activity of I3C6. When glucose was co-incubated with maltoheptaose, a dramatic reduction in glucose was observed with a simultaneous increase in maltose, but there was no detectable change in maltoheptaose (Appendix A). While combined incubation with fucose and maltoheptaose showed no significant change in fucose (eluting around 2 min), maltoheptaose was degraded to smaller maltooligomers, including maltose but not glucose (Appendix A). Given I3C6’s modest activity on starch and the lack of degradation of the cyclodextrins, our investigation suggests that this enzyme has little endo activity but a preference for degradation of maltoheptaose in the absence of glucose.

In summary, the three enzymes showed distinct profiles towards different carbohydrate substrates, with variable levels of hydrolytic activity towards starch, cyclodextrins, and maltooligosaccharides, and only Ika2 showing evidence for transglycosylation activity.

### 3.3. Ika2 Forms a Stable Dimer, While Rho13 and I3C6 Are Mainly Monomeric

The three enzymes showed different elution behaviors on a size exclusion column (Figure 2). Rho13 eluted with two major peaks, both of which corresponded to the pure protein according to SDS-PAGE. Both peaks could be fitted to single Gaussian curves (Figure 2a). Based on calibration of the column with proteins of known size, the first peak had a retention volume corresponding to a dimer, and the second that of a monomer (Appendix A). The dimer and monomer peaks corresponded to 26% and 74% of the combined peak area respectively, and both peaks showed significant enzymatic activity (based on measuring the formation of reducing ends in an MBTH assay [74]). Ika2 eluted as a single peak, though this was slightly skewed (Figure 2b). The observed peak top around 13 mL predicted a molecular weight of 92 kDa, lying between the monomeric and dimeric values of 69 and 138 kDa, respectively. This suggests a dynamic equilibrium between monomer and dimer, which exchange too rapidly to elute as separate peaks. I3C6 also showed a major peak preceded by a broad but low plateau (Figure 2c). Only the major peak (which corresponded to 84% of the total elution area) showed significant activity. The peaks eluting prior to the main I3C6 peak were smaller, but bands corresponding to monomeric I3C6 were reproducibly observed by SDS-PAGE in the corresponding lanes (Figure 2c), and we therefore attributed them to nonspecific aggregates or oligomers of I3C6 that dissociated during SDS-PAGE.

To elucidate the oligomerization states of the proteins more directly, we recorded the SAXS curves of the three enzymes (Figure 4a, Table 1). For I3C6, the radius of gyration (*R*_g_) increased with protein concentration (Figure 4b). We attribute this to nonspecific aggregation, since the SAXS data could not be fitted to a mix of monomers and dimers. In contrast, the Rho13 data were best fit by a combination of the AlphaFold-predicted structure for the monomer and the symmetry-built dimer. Furthermore, all concentrations could be described using the linear combinations of the models with some deviation at higher concentrations, suggesting a dimerization equilibrium of Rho13 (Figure 4b), as expected from the SEC data. The SAXS curves for Ika2 were fitted using an AlphaFold-predicted Ika2 dimer. This led to a value of R_g_ independent of concentration and consistent with the R_g_ calculated for a dimer. Dimerization is a common feature for many of Ika2’s closest homologs (54–57% identity): amylase from *Thermus* sp. (PDB: 1SMA) is a dimer [65], cyclomaltodextrinase from *Bacillus* sp. (PDB: 1EA9) is a dodecameric hexamer of dimers [13], and neopullulanase from Geobacillus stearothermophilus (PDB: 1J0H) is also a dimer [48]. In addition, 8 of the 10 published structures in GH13_20 form dimers.

### 3.4. Activity Profiles as a Function of Temperature, pH, and Ca^2+^

Enzymatic activity (hydrolysis of starch) under a range of different conditions was determined using the MBTH assay, which measures formation of reducing carbohydrate ends. The organisms from which the three enzymes were obtained all live in a cold environment, that is a mixture of seawater and alkaline spring water, making it reasonable to expect that they would prefer an alkaline and modestly saline environment with a low optimal temperature. We therefore examined the impact of temperature, pH, and salt.

#### 3.4.1. Temperature

All three enzymes were preincubated at a given temperature for 15 min, after which product formation was measured from a linear increase in product signal over a 5.5 min period (see typical examples in Appendix A). Measured in terms of specific activity (moles reducing ends formed per minute per mole enzyme) at pH 7.5, Rho13 is an order of magnitude more active than the two other enzymes (Figure 5a): on average 30- and 15-fold more active than Ika2 and I3C6, respectively. However, when normalized to each enzyme’s maximal activity, the thermal profiles of the three amylases were similar, with optima of 30–35 °C (Figure 5b) and significant activity below 20 °C. Ika2 showed particularly pronounced psychrophilic properties, with 50% maximal activity at 5 °C, as well as being the most thermostable, with 40% activity remaining at 55 °C. However, all three enzymes can be classified as psychrophiles, with significant activity below 20 °C.

##### 3.4.2. pH Profile

Rho13 and I3C6 showed a clear preference for higher pH, with optimal activity around pH 8–9. However, they differed in their alkaline sensitivity: I3C6 was completely inactive at pH 9.5, while Rho13 still showed 78% activity at this pH. In contrast, Ika2 was markedly acidophilic (despite its origin in ikaite columns), with a marked decline in activity from pH 6 up to pH 8 (Figure 5c). At pH 6, Ika2 activity was increased ca. 3.3-fold compared to at pH 7.5, while that of Rho13 decreased ca. 7.5-fold, making Ika2 ~1.5-fold more active than Rho13 at pH 6, while the absolute activity of Rho13 at its optimal pH of 8.0 was still at least 7-fold higher than that of Ika2 at pH 6. Thus, Rho13 is overall significantly more active than the two other enzymes.

##### 3.4.3. Salt Profile

All three enzymes are reasonably halotolerant, though Ika2 and I3C6 showed some decline in activity at >150 mM NaCl (Figure 5d).

In summary, the three enzymes all showed significant activity at low temperatures, peaking slightly above ambient temperatures, but showed individual variations in terms of sensitivity to pH and ionic strength. Ika2 had the broadest temperature profile and only moderate dependence on salt, but its preference for low pH makes it less appealing as a detergent enzyme, given the preference for alkaline conditions in, for example, laundry washing [21]. Here Rho13 has the advantage of a strongly alkaliphilic profile, as well as a robust indifference to ionic strength, though its activity declines more steeply with lowered temperature than does that of Ika2. I3C6 has the weakest attributes, with low specific activity, complete abolition of activity at > pH 9.0, and a slow decline in activity with ionic strength. As a final differentiator of activity, we turned to the impact of Ca^2+^ on enzyme activity.

##### 3.4.4. Analysis of Ca^2+^ Binding of Ika2, Rho13, and I3C6

As we will describe in detail below, there is good reason to expect Ca^2+^ to bind to all three enzymes and affect their activity and stability. Accordingly, in the following section we describe the experimental effect of Ca^2+^ ions on the activity, structure, and stability of the three enzymes and then examine the structural basis for the role of Ca^2+^ based on computational analysis. Activity was measured by MBTH assay (Figure 5e), structure by far-UV circular dichroism (CiD, Figure 6a), and thermal stability either through changes in the proteins’ secondary structure by CiD (Figure 6b) or tertiary structure by differential scanning fluorometry (DSF) (Figure 6c). CiD data could in all cases be fitted with a three-state model, involving the native state N, a partially denatured intermediate state I, and a denatured state D. Where possible, data from 21 different wavelengths (210–230 nm in steps of 1 nm) or 9 different wavelengths (Ika2 with no EDTA) were fitted globally to obtain the melting temperatures for the two transitions (N-to-I and I-to-D). The results are summarized in Table 2 and described in detail below.

###### Ca^2+^ Has Little Effect on Ika2 Activity, but Rho13 and I3C6 Follow a Bell-Shaped Dependence

Ika2 showed no increase in activity upon the addition of Ca^2+^ up to 20 mM, but rather a slight decline (Figure 5e). For this reason, activity was determined in the presence of EDTA, which had no significant influence on activity, suggesting that Ca^2+^ has no positive effect on activity (Appendix A, time profiles in Appendix A). In contrast to Ika2, the addition of Ca^2+^ to both Rho13 and I3C6 shows a bell-shaped curve with an activity optimum around 2.5 mM CaCl_2_ (Figure 5e). The positive effect of Ca^2+^ on activity can be seen directly in the time profiles when adding either EDTA or CaCl_2_ at different time points (Appendix A). The activity data can formally be fitted to a model in which the first, somewhat weak, binding step (inflection point ca. 0.5 mM) increases activity and the second, even weaker, binding step (inflection point ca. 8–20 mM) decreases it. The decline in activity at higher Ca^2+^ concentrations is indicative of nonspecific inhibition, seen for numerous other enzymes. In the case of Rho13, we speculate that this second binding event might involve consensus sequence 2 (CS2, see below), since Ca^2+^ binding here would increase rigidity and lower activity.

###### Removal of Ca^2+^ Affects Enzyme Structure and Reduces Stability

Rho13

Thermal denaturation by far-UV CiD in the absence of EDTA showed a gradual redshift from a 222 nm peak to a flatter curve from around 53 °C onwards (Figure 6b). The shift was fitted in OriginPro2024 to a three-state unfolding model, with the first transition corresponding to a global denaturation and the second interpreted to nonspecific aggregation. This first unfolding step showed a *t_m_* of 55.98 ± 0.04 °C, and the second 58.32 ± 0.14 °C. Consistent with this, DSF showed a transition of 55.215 ± 0.007 °C, with no reversibility of unfolding (Appendix A). The *t_m_* was significantly above the temperature optimum of 35 °C (Figure 5b). Activity declined steeply above 45 °C. This suggests local unfolding of the enzyme that presumably involves the active site, thus leading to the thermal inactivation of Rho13 observed in the MBTH assays. There was no significant change in the far-UV CiD structure of Rho13 with or without EDTA (Figure 6a), but the enzyme showed a significantly changed CiD unfolding curve (Figure 6b), suggesting the existence of a structurally important Ca^2+^ binding site. Consistent with this, DSF showed that EDTA lowered the t_m_ by 2 °C.

Ika2

While EDTA did not affect Ika2’s activity (Figure 5e and Appendix A), it significantly altered the enzyme’s secondary structure according to far-UV CiD (Figure 6a). The major minimum around 208 nm was redshifted, and the intensity was amplified. This suggests a Ca^2+^ binding site, which is structurally important, as seen in the effect on stability, but with a deleterious effect on activity above 1 mM CaCl_2_ (Figure 5e). The thermal denaturation data could not be fitted with great confidence due to visible aggregation at these high temperatures. Nevertheless, an acceptable fit was made, predicting two consecutive *t_m_* values of 57.2 °C and 62.1 °C in 2 mM CaCl_2_ (Table 2), which change to 47.8 °C and 72.2 °C in EDTA; the second transition is likely aggregation. DSF, which monitors changes in Trp fluorescence and thus reports on the global structure, showed a single melting transition with a midpoint of 62.0 °C, which fell dramatically to 50.8 °C in EDTA (Figure 6b and Table 2). Thus, in CaCl_2_, the secondary and tertiary structure are both lost around 57–62 °C, well above the optimal activity temperature of 30 °C.

I3C6

Thermal unfolding of I3C6 as measured by CiD and DSF showed *t*_m_ values of 52.47 °C (first transition) and 55.40 °C, respectively (Figure 6b,c). EDTA had a small effect on the far-UV CiD spectrum, which is likely due to concentration differences and does not show any spectral shifts (Figure 6a). Surprisingly, the CiD-measured *t*_m_ increased by 3 °C to 55.7 ± 0.25 °C upon the addition of EDTA. This is likely not significant, as the DSF-based *t*_m_ values with and without EDTA were reversed compared to those seen with CiD (Table 2).

In summary, the three enzymes differ significantly in their dependence on Ca^2+^ for stability and activity, and the two phenomena are not correlated. Ika2’s activity is unaffected by Ca^2+^, but its removal markedly destabilizes the enzyme; conversely, I3C6 and Rho13 both show optimal activity around 2.5 mM Ca^2+^, but are destabilized to very different extents. Clearly, they must differ considerably in their interactions with Ca^2+^.

### 3.5. Computational Analysis of Ca^2+^ Binding to the Three Enzymes

Having shown that Ca^2+^ has a profound but complex effect on the three amylases, we now turn to computational analysis of the potential interactions involved.

#### 3.5.1. Ika2 Shows No Evidence of Stable Ca^2+^ Binding

GH13_20 enzymes such as Ika2 often have a metal binding site. Five PDB crystal structures from this family show a bound Ca^2+^ and share high sequence preservation in this region (Figure 7a), which involves the amino acid motif NGDPSND---GGD (with metal ion coordination by the Asp residues). The two closest homologs of Ika2 (1GVI and 1EA9) have no assigned Ca^2+^ in their crystal structure (possibly due to loss during purification and crystallization), though they retain this Ca^2+^ binding motif. We note that the Ca^2+^ binding site is not completely conserved in the GH13_20 family. Thus, 2Z1K, 2WC7, and 4AEF have mutations among the coordinating residues; in addition, in 4AEF the sequence Q224-Y233 is not modelled, indicating a lack of metal coordination. Given Ika2’s almost complete conservation of the Ca^2+^ binding motif (Figure 7a), it would be expected to bind Ca^2+^; nevertheless, Ca^2+^ had hardly any effect on Ika2 activity (Figure 5e). MD simulations corroborated these observations. They showed only an increase in the average RMSD and RMSF values of 0.2 Å and 0.1 Å, respectively, in 1.2 μs trajectories conducted with and without Ca^2+^. This suggests that the absence of Ca^2+^ at best slightly increases the flexibility of Ika2. As summarized in Appendix A, a comparative analysis of the per-residue RMSF values revealed that only 24 residues, representing 2% of the dimer, exhibited increased fluctuations above the threshold of 0.5 Å in the trajectory without Ca^2+^. Moreover, half of those residues are located at or near the Ca^2+^-binding sites, and only five are located near the active site cleft, indicating that the absence of Ca^2+^ should not have a major impact on residues involved in substrate interactions. This further supports our experimental finding that the absence of Ca^2+^ does not have a negative impact on catalytic activity (Figure 5e). Similar observations were noted in the Ika2 homolog neopullulanase (PDB code 1J0H), where Ca^2+^ is proposed to be non-essential for catalysis [48].

#### 3.5.2. Rho13 Likely Has One Ca^2+^ Binding Site

Solved structures of homologs of Rho13 in the GH family 13_36 have varying numbers of preserved Ca^2+^ binding sites. To the best of our knowledge, Rho13 has two consensus sequences (CSs): CS1 (DSDGDGTGD) (Figure 7b) and CS2 (VINH---PDL---(HI)/(YF)), identified in 3K8K, 7JJT, and 5M99, at the interface of three loops close to the active site. The α-amylase from *Halothermothrix orenii* (PDB: 1WZA), Rho13’s closest homolog, shares these two consensus sequences, but only CS1 is occupied by Ca^2+^. A second Ca^2+^ in 1WZA is lodged in a second Ca^2+^ binding loop (residues 65–77), a structural element that is also observed in 7JJT, but that binding loop does not exist in Rho13’s structure. Therefore, our computational structure of Rho13, used in subsequent MD simulations, contains a single Ca^2+^ ion (Figure 1a), which was transferred from the 1WZA homolog to the first conserved Ca^2+^ binding site of Rho13 after a structural alignment of both structures.

#### 3.5.3. I3C6 Only Maintains One Stably Bound Ca^2+^ Ion

I3C6 was previously reported to have increased activity when CaCl_2_ is added [27], and several I3C6 homologs in the GH family 13_5 have been shown to contain multiple binding sites. The amylase from *Bacillus stearothermophilus* (PDB: 1HVX) has a generally preserved Ca^2+^(I)-Na^+^-Ca^2+^(II) binding site with three ions in a line, and a third Ca^2+^(III) binding site is located at the interface of two structural domains. Accordingly, we computationally transferred the Ca^2+^(I)-Na^+^-Ca^2+^(II) triad and the Ca^2+^(III) ion from the 1HVX homolog to the I3C6 structure predicted by AlphaFold (Appendix A) and conducted MD simulations to better understand the Ca^2+^ binding of this psychrophilic enzyme. Interestingly, our simulations revealed that only Ca^2+^(I) and Na^+^ ions remained in their binding sites, whereas Ca^2+^(II) and Ca^2+^(III) abandoned their original positions throughout the simulations (Appendix A). This may have resulted from existing differences in the amino acid composition of the Ca^2+^ binding sites between the psychrophilic enzyme and its homolog. While the Ca^2+^(I)-Na^+^-Ca^2+^(II) binding site of 1HVX exhibits seven Asp residues (D105, D162, D186, D197, D203, D205, and D207), the same region of I3C6 only presents three (D195, D201, and D203) (Appendix A). Consequently, the number of ionic interactions established between the ions and the residues of the binding site is much smaller in I3C6, resulting in looser binding of the Ca^2+^(I)-Na^+^-Ca^2+^(II) triad. Among the three cations, Ca^2+^(II) established fewest interactions with I3C6 and was most exposed to the solvent, thus explaining its rapid release from the binding site in all simulation attempts. This may also explain the escape of Ca^2+^(III) from its initial position in the MD simulations conducted with I3C6. The region surrounding the transferred Ca^2+^(III) ion contains fewer Asp residues (one in I3C6 vs. two in 1HVX) and includes two positive amino acids, K405 and R429, that correspond to S406 and G431 in the 1HVX homolog, respectively (Appendix A). The increase in the positive character of this region makes it less likely to accommodate the Ca^2+^(III) which, without good binding contacts, easily leaves the binding site. In sum, the simulations suggest that only a single Ca^2+^ ion can persistently bind the I3C6 structure, and this was used for all subsequent MD runs performed with this psychrophilic enzyme.

### 3.6. Psychrophilic Features of the Three Amylases Analyzed by MD Simulations

Psychrophilic enzymes exhibit distinct structural and dynamic features that allow them to operate effectively in cold environments [2]. Compared to their mesophilic homologs, psychrophilic enzymes are characterized by increased structural flexibility, which is often linked to a lower number of Pro residues, fewer ionic and polar interactions, an increased presence of Gly residues, and enhanced solvent accessibility [75]. Collectively, these factors reduce the energy barriers for molecular movement at lower temperatures and influence activation free energy by respectively decreasing and increasing enthalpic and entropic contributions. According to transition state theory, this adjustment is advantageous, as enthalpy-driven reactions decrease exponentially with temperature reduction, while entropic contributions remain unaffected by temperature changes [76,77]. Nonetheless, excessive flexibility in regions unrelated to catalysis will not alter the enthalpy term, but will elevate the entropic barrier. This can be mitigated either by increasing local rigidity in noncritical regions that do not affect catalytic movements or by concentrating enhanced flexibility in areas relevant to catalysis (albeit not necessarily at the active site) [76,77].

To better understand the conformational dynamics of the Rho13, Ika2, and I3C6 psychrophilic amylases, we conducted MD simulations with these three enzymes and with their respective thermophilic homologs (1WZA, 1HVX, and 1J0H). We evaluated the RMSD of backbone atoms along the concatenated 1.2 μs trajectory for the six amylases, using the minimized structure of each enzyme as a reference. The average RMSD values for the psychrophilic enzymes consistently exceeded those observed for the respective mesophilic counterparts (Table 3), indicating that the cold-active enzymes are less stable than their corresponding homologs. This aligns with what is hypothesized in the literature [75,78]. Specifically, the I3C6 vs. 1HVX pair showed the largest variance (0.6 Å), while Rho13 vs. 1WZA displayed the smallest difference (0.2 Å). Additionally, the standard deviation of the presented RMSD values was consistently higher in the psychrophilic enzymes, suggesting more pronounced structural fluctuations.

We performed a structural comparison of the psychrophilic enzymes and their mesophilic homologs by evaluating the *R_g_*, SASA, and intra-protein hydrogen bond count throughout the concatenated 1.2 μs simulations. Our findings revealed a lower average number of intra-protein hydrogen bonds and an increased average *R_g_* value for Rho13 and I3C6 (Table 3), suggesting a weakening of stabilizing interactions and overall less compact structures compared to their mesophilic counterparts. In contrast, Ika2 showed similar values for hydrogen bonds and *R_g_* relative to its homolog, potentially attributable to its dimeric nature, which may influence these metrics. Nonetheless, Ika2 exhibited a 10% decrease in the average number of salt bridges compared to its mesophilic counterpart, which also points to a disruption of stabilizing interactions. All three psychrophilic enzymes demonstrated higher average SASA values (Table 3), underscoring a trend toward decreased compactness and improved interactions with the solvent.

RMSF values (i.e., levels of structural changes during MD simulations) for the three psychrophilic enzymes and their mesophilic counterparts reinforced our findings on psychrophilic conformational dynamics (Appendix A). The RMSF values of the backbone consistently indicated an overall increase in the structural flexibility of psychrophilic enzymes compared to the mesophilic variants (Table 3). However, drawing significant conclusions from average RMSF values can be challenging. To gain deeper insights, we delved into residue-specific flexibility by comparing the RSMF values of corresponding residues between each psychrophilic enzyme and its mesophilic counterpart, using a ±0.5 Å threshold to identify notable changes in the flexibility of each residue. The cold-adapted amylases exhibited a larger number of residues exceeding this threshold (Figure 8a–c, represented by magenta sticks), indicating that larger portions of the psychrophilic enzymes have increased flexibility relative to their mesophilic homologs.

Ika2 displayed 111 residues with at least a 0.5 Å increase in RMSF values compared to their corresponding residues in the 1J0H homolog (Figure 8b), constituting approximately 10% of the enzyme, which indicates enhanced global flexibility. Conversely, 16 residues exhibited a >0.5 Å decrease in flexibility, indicating targeted increases in rigidity. Interestingly, the flexibility distribution varied between the two chains of the enzyme. Among the residues with increased flexibility, 76 were located on chain A and 35 on chain B, whereas increased rigidity was more pronounced in chain B with 13 residues, compared to only 3 in chain A. A domain-specific analysis revealed that 40% of the residues with increased flexibility are located within domain N (Figure 8b), which is crucial for defining the active site cleft and is known to undergo substantial structural rearrangements upon substrate binding in the homolog neopullulanase [48]. This suggests a potential role for this domain in enhancing substrate interaction at low temperatures. Domain A, which contains the common (β/α)_8_-barrel motif found in amylase family enzymes, and together with domain N forms the active site cleft, accounted for 32% of the residues with increased flexibility (Figure 8b). Conversely, domain B had the lowest percentage of residues with increased flexibility (2%) and the highest percentage (56%) of residues with increased rigidity (i.e., residues whose RMSF is >0.5 Å below the RMSF of the equivalent residues of the mesophilic homolog enzyme—Figure 8b, represented by yellow sticks), underscoring its potential role in maintaining structural integrity near the catalytic site. Domain C, which is the farthest from the active site and features solvent-exposed mobile loops, exhibited 27% of residues with increased flexibility. Of note, 17 of the 111 residues that showed increased flexibility are in important positions for substrate recognition and interaction, as identified in substrate–enzyme complex structures of the homolog (PDB codes 1J0I, 1J0J, and 1J0K). Notably, none of the catalytic residues in Ika2, i.e., the critical D330/E359/D426 triad and others promoting relevant ionic and polar interactions for catalysis (Y209, H249, R328, H425, D470, and R474), showed increased flexibility, in accord with the literature [75,77].

Rho13 and I3C6 have 21 and 30 residues, respectively, with increased flexibility compared to the matching 1WZA and 1HVX homologs (Figure 8a,c, represented by magenta sticks). Most of these residues are in domain A (nearly 57% of all residues with increased flexibility), specifically in the region that forms the active site cleft, as also observed for the Ika2 amylase. We believe that the enhanced dynamics of this area may give the psychrophilic enzymes easier access to the substrate and/or facilitate product release, and possibly enable them to accommodate substrates with alternative dimensions and/or orientations (i.e., broader substrate specificity), which are all features commonly associated with cold-adapted enzymes [61,75,78]. Interestingly, while in Rho13 the remaining 43% of the highly flexible residues are found in domain B (Figure 8a), I3C6 displays the same percentage of highly flexible residues in domain C (Figure 8c). This clearly indicates that, although common trends can be observed among the three psychrophilic amylases studied, each enzyme seems to have developed its own strategy to improve its global/local flexibility and preserve its catalytic activity in cold environments.

The RMSF analysis highlights increased global and local flexibilities on the three cold-adapted amylases vs. their mesophilic homologs. These enhanced local flexibilities are mostly concentrated in regions that should play pivotal roles in directing the substrate pathway toward the active site. In addition, other regions with increased flexibility were also found in peripheral areas of the psychrophilic enzymes, which is consistent with a previous study that hypothesized that enhanced flexibility in non-catalytic regions is a common adaptation of psychrophilic enzymes [79]. Interestingly, both Rho13 and Ika2 exhibited some residues with reduced flexibility compared to their mesophilic counterparts (Figure 8a,b, represented by yellow sticks). This increased rigidity in specific regions likely aids in reducing the entropy barrier for catalysis, highlighting a delicate equilibrium between flexibility and rigidity that may enhance enzyme efficiency in cold climates, as discussed in the literature [76,80,81].

To elucidate the largest-amplitude collective motions and reduce dimensionality, we performed PCA by diagonalizing the backbone covariance matrices of the concatenated trajectories. Analysis of the first 50 eigenvectors, which together account for over 90% of the total molecular motions, revealed that the eigenvalues are consistently higher in psychrophilic enzymes than in their mesophilic counterparts. This observation suggests that more significant conformational changes occur in the psychrophilic enzymes. By projecting the trajectory onto the two principal eigenvectors, which represent approximately 50% of the overall motions and delineate a critical subspace for observing the most significant collective motions, we observed that psychrophilic enzymes explore a broader area of conformational space than their mesophilic homologs (Figure 8d–f). This increased exploration implies a higher degree of conformational flexibility, likely facilitated by the easier transition over energy barriers among multiple local minima. Such behavior in psychrophilic enzymes could be attributed to reduced structural stabilizing interactions and compactness, as discussed above.

## 4. Conclusions

We have isolated and characterized three novel secreted amylases, which, despite their common origin in an alkaline and Ca^2+^-rich environment in Greenland’s ikaite columns, show significant differences in terms of optimal pH and Ca^2+^-binding conditions, as well as in their ability to hydrolyze starch and other carbohydrates. Nevertheless, they share the psychrophilic property of losing activity at temperatures well below those where there is detectable loss in structural integrity. We attribute this to the high levels of conformational flexibility, which facilitate easy binding and accommodation of substrate molecules and exploration of the catalytic energy landscape, but at the expense of structural stability. This is reflected in the reduction of internal polar interactions such as hydrogen bonds and salt bridges, consistent with the general psychrophilic strategy of providing cold activity through conformational flexibility.

## Figures and Tables

**Figure 1 biomolecules-15-00415-f001:**
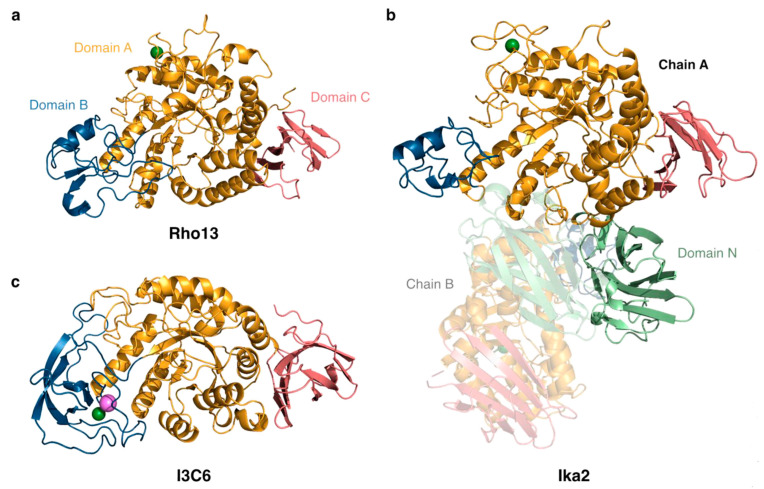
Cartoon representations of the CoLabFold/AlphaFold-predicted structures of the (**a**) Rho13, (**b**) Ika2, and (**c**) I3C6 psychrophilic α-amylases. Domains A, B, C, and N are shown in bright orange, iridium blue, salmon pink, and light green, respectively. Ca^2+^ and Na^+^ ions are depicted as green and violet spheres, respectively.

**Figure 2 biomolecules-15-00415-f002:**
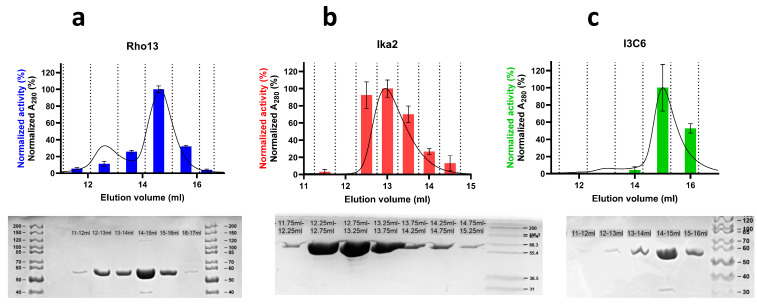
Oligomer distribution and activity of Rho13 (**a**), Ika2 (**b**), and I3C6 (**c**). Ni-NTA–eluted proteins were subjected to SEC (elution profiles shown as joined black lines), and fractions were tested for activity using an MBTH assay (columns with standard error of linear fit as error bars). Activity was detected in both Rho13 peaks, the last I3C6 peak, and throughout the skewed Ika2 peak. SDS-PAGE showed single-band purity and similar mobility of all fractions for the same protein, indicating that multiple SEC peaks are oligomers. Original images of (**a**–**c**) can be found in Appendix A.

**Figure 3 biomolecules-15-00415-f003:**
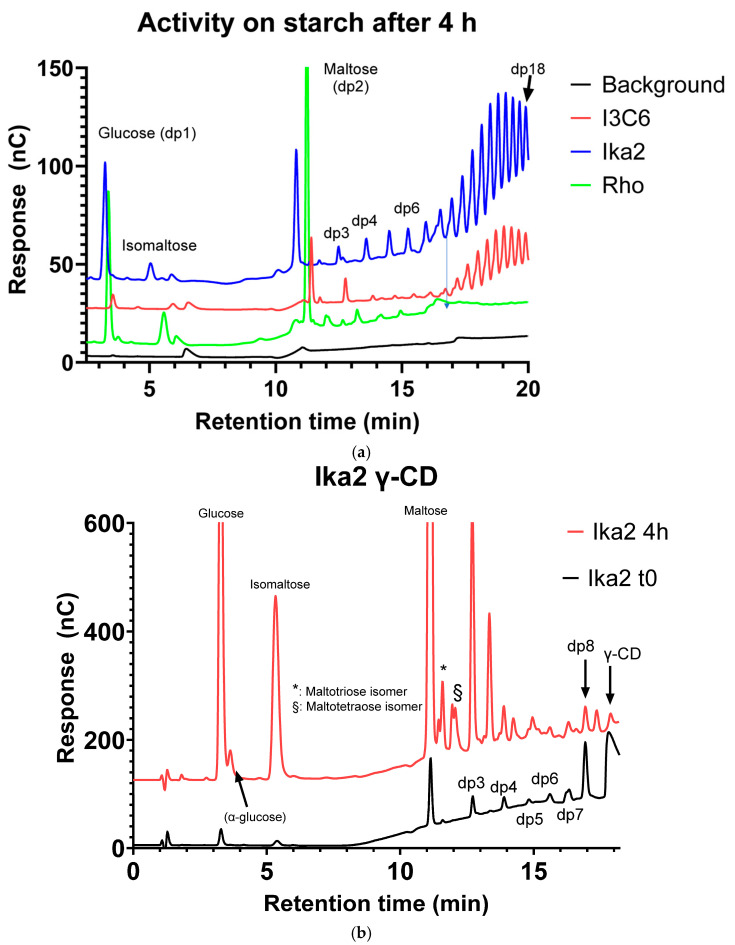
Degradation and modification of carbohydrates by the three amylases analyzed by pulsed amperometric detection. Products produced by (**a**) all three amylases upon treatment with starch and (**b**) Ika2 incubated with γ-cyclodextrin. (**c**) Transglycosylation of maltoheptaose in the presence of fucose carried out by I3C6. Additional elution profiles are shown in Appendix A.

**Figure 4 biomolecules-15-00415-f004:**
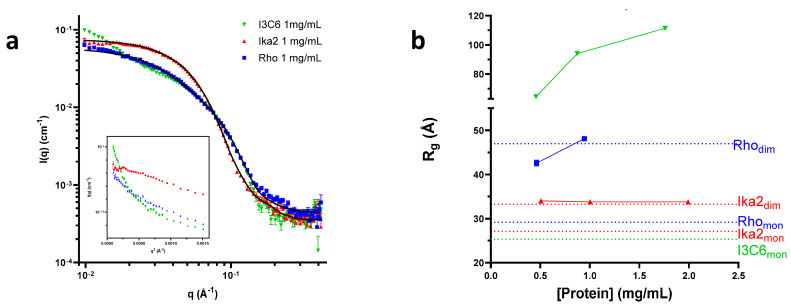
SAXS analysis of the quaternary structure of the three amylases. (**a**) Small angle x-ray scattering profiles of 0.5, 1, and 2 mg/mL enzyme were fitted to AlphaFold predictions of the protein structure using SUPERSAXS. The panel shows scattering profiles at 1 mg/mL enzyme. (**b**) The SAXS data allowed determination of the radius of gyration, R_g_. R_g_^measured^ (data points) was compared to R_g_^calculated^ (stippled lines) for dimers and monomers. The data are consistent with a dimer for Ika2 and a mix of monomers and dimers for Rho13. Ika2 (red pyramids) was very well described by the dimer, while Rho13 (blue squares) radius was best estimated with a mix of monomers and dimers; only 0.5 and 1 mg/mL datasets were obtained for Rho13 due to aggregation. I3C6 (green triangles) showed a concentration-dependent size increase, which we attributed to aggregation, as seen by the increase in scattering at low *q*-values (panel a insert: Guinier plot).

**Figure 5 biomolecules-15-00415-f005:**
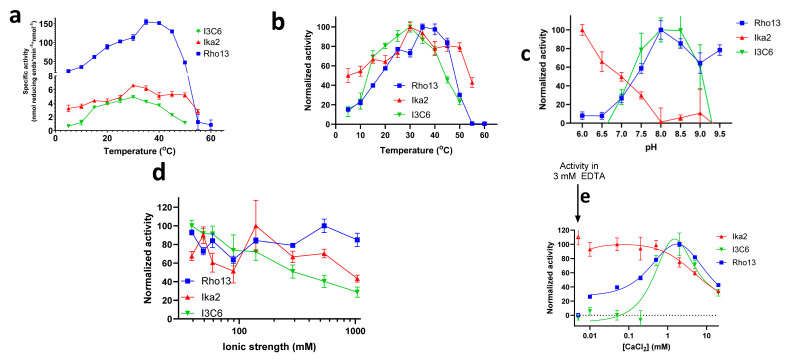
Activity optima. Absolute specific activities and relative activities were measured by MBTH assay on the breakdown of starch and formation of reducing ends at different buffer conditions. Rho13 (blue squares), Ika2 (red pyramids), and I3C6 (green triangles). (**a**) Specific activities measured at the given temperature at pH 7.5 after 10 min of previous incubation. (**b**) Temperature data normalized to the maximal activity for each enzyme. All enzymes show a clear psychrophilic profile, with low t_opt_ and activity below 20 °C. (**c**) pH optimum. Rho13 and I3C6 showed highest activity under alkaline conditions while Ika2 preferred acidic conditions. (**d**) Effect of ionic strength (0–1000 mM NaCl together with 2 mM CaCl_2_ and 50 mM MOPS buffer). The enzymes were relatively unaffected by increasing ionic strength, though with a slight declining trend for I3C6. (**e**) CaCl_2_ optima. The hollow symbols in left side of graph are activity in presence of 3 mM EDTA relative to 2 mM CaCl_2_. Rho13 and I3C6 had a defined optimum at 2 mM, while Ika2 had the highest activity at 0.01 mM. Additional CaCl_2_ inhibited the activity. The activity was fitted to Equation (1) (solid lines). All data are initial velocities over 330 s, with error bars as standard error of a linear fit.

**Figure 6 biomolecules-15-00415-f006:**
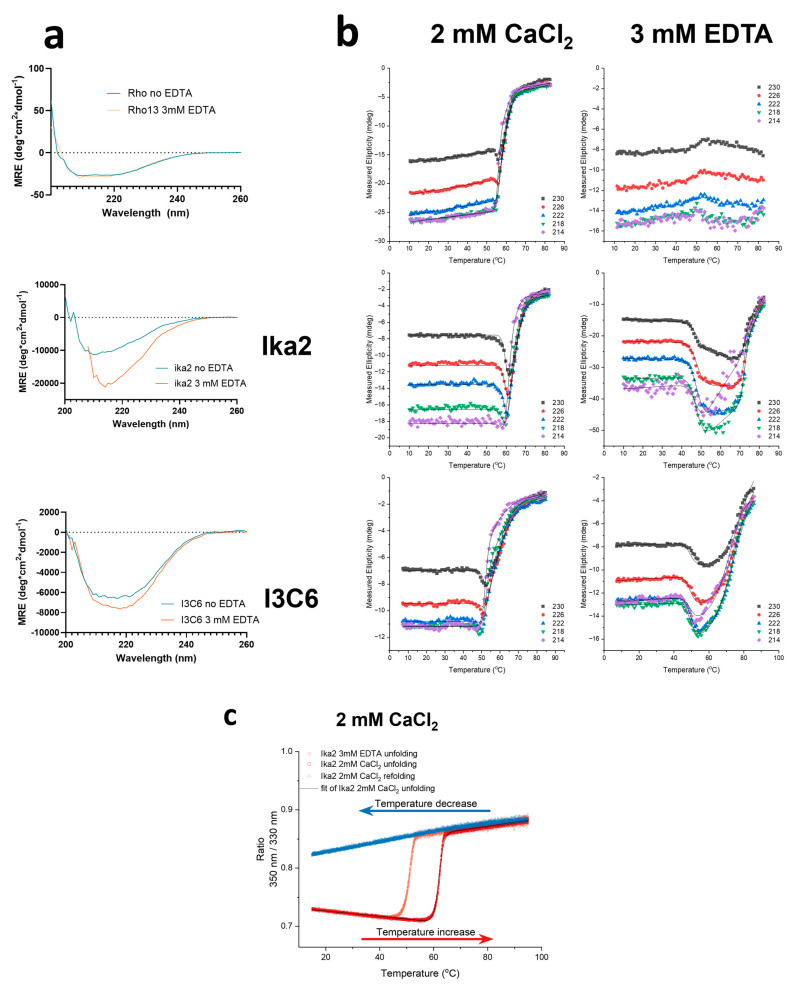
Structure and stability of the three amylases, as measured by spectroscopy. (**a**) CD spectra of enzymes with/without 3 mM EDTA (**b**) Thermal stability of amylases in the presence or absence of 3 mM EDTA. Enzymes were cooled from room temperature to 10 °C and then heated at a scan rate of 1 °C/min, while full CD spectra were recorded (200–260 nm). Top: Selected traces monitored at different wavelengths in the α-helix region (214–230 nm) are shown (points), along with the unfolding fit (solid lines). (**c**) Example data of thermal unfolding, as measured by differential scanning fluorometry. Amylases were slowly heated from 15 to 95 °C at 1 °C/min (red squares and orange circles) and recooled to measure refolding (blue triangles), and the unfolding regime was fitted to a two-state unfolding (line). Red squares show the unfolding of Ika2 at 2 mM CaCl_2_, and orange circles show the unfolding of Ika2 with 2 mM CaCl_2_ and 3mM EDTA. Data are the overlay of three technical repeats measured in parallel.

**Figure 7 biomolecules-15-00415-f007:**
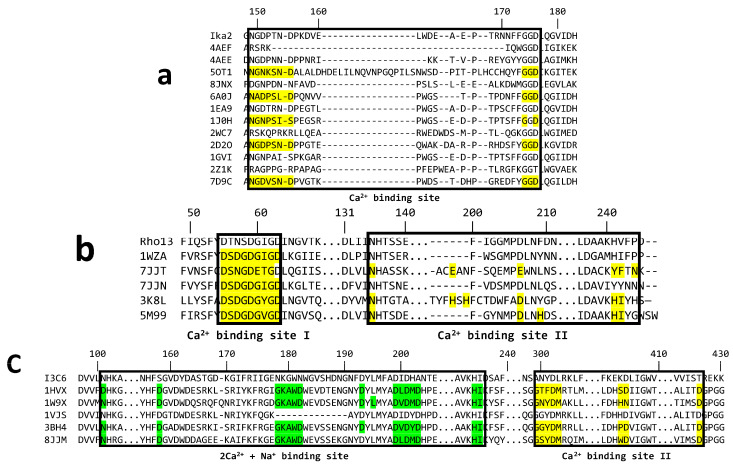
Sequence alignment with of calcium binding sites from Protein Data Bank (PDB), performed using T-Coffee. One sequence representing the different enzymes in the CAZy (a: GH13_20, b: GH13_36, c: GH13_5) database with structures submitted. (**a**) Ika2 aligned with 12 GH13_20 enzymes with structures in CAZy. The numbers on top correspond to the Ika2 sequence. A single binding site is present in five structures; in these, residues within 5Å are colored yellow. (**b**) Rho13 aligned with 5 GH13_36 structures with metals in their submitted structures. “…” denotes hidden parts of the alignment. The numbers on top correspond to the Rho13 sequence, including the signal peptide. Two sites seem to be conserved in Rho13, as well as a loop that is present in all the other GH13_36 structures, and possibly a second site that is occupied by calcium in three of the structures, but is conserved in all. (**c**) I3C6 aligned with GH13_5 amylases from CAZy. Green represents residues (backbone or sidechain) coordinating the Ca-Na-Ca binding site present in these enzymes, which are mostly conserved in I3C6. Yellow represents a second conserved binding site that is imperfectly conserved in I3C6.

**Figure 8 biomolecules-15-00415-f008:**
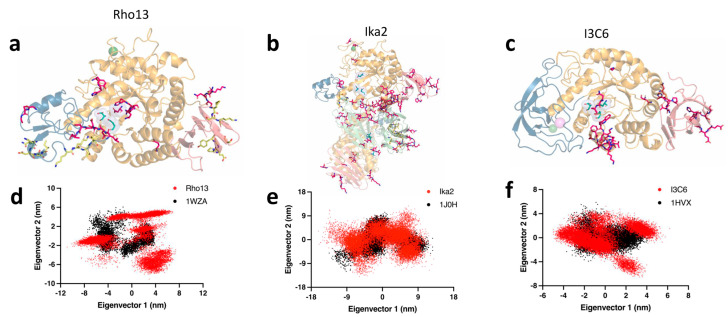
Compilation of the results concerning the MD simulations performed with the three psychrophilic α-amylases (Rho13, Ika2, and I3C6) and their respective homologs (1WZA, 1J0H, and 1HVX). (**a**–**c**) Cartoon representations of the Rho13 (**a**), Ika2 (**b**), and I3C6 (**c**) enzymes that highlight the residues that, throughout the MD simulations, exhibited increased flexibility (represented by magenta sticks) or increased rigidity (represented by yellow sticks) compared to the corresponding homologs. The domains A, B, C, and N are shown in bright orange, iridium blue, salmon pink, and light green, respectively. The catalytic D/E/D triads are represented by light blue sticks. The calcium and sodium ions are depicted as green and violet spheres, respectively. (**d**–**f**) Projection of the concatenated 1.2 μs MD trajectories onto the two principal eigenvectors (eigenvectors 1 and 2, which represent approximately 50% of the overall motions) for each pair of psychrophilic (red dots)/homolog (black dots) enzymes.

**Table 1 biomolecules-15-00415-t001:** Oligomerization properties of the three enzymes probed by SAXS.

Protein	Concentration_Abs280_ (mg/mL)	R_g_ (Å) ^a^	Model	χ^2 b^	Concentration_SAXS_ (mg/mL)	n-mers ^a^ (SAXS ^b^)
Ika2	1.99	33.8 ± 0.1	AF dimer	4.3	1.69	1.67 (1.97)
	1.00	33.8 ± 0.1	AF dimer	1.9	0.84	1.67 (1.99)
	0.51	34 ± 0.2	AF dimer	1.4	0.42	1.63 (1.96)
Rho	0.94	49.8 ± 0.5	Monomer_0.03 mg/mL_ + dimer_0.69 mg/mL_	2.9	0.72	1.58 (1.96)
	0.46	42.6 ± 0.7	Monomer_0.06 mg/mL_ + dimer_0.29 mg/mL_	1.4	0.35	1.38 (1.83)
I3C6	1.76	64.5 ± 1	-	-	-	1.6
	0.87	94 ± 0.8	-	-	-	3.42
	0.46	111.4 ± 0.4	-	-	-	7.4

Notes: ^a^ Determined from the indirect Fourier transformation routine, as described in the Section 2. ^b^ SAXS modelling results using the described model. Protein concentrations were obtained from the model fit using an absolute scale.

**Table 2 biomolecules-15-00415-t002:** Optimal activity temperatures and melting temperatures (*t*_m_) obtained from the thermal unfolding of the three enzymes Ika2, Rho13, and I3C6, as monitored by either DSF or far-UV CiD in the presence of 2 mM CaCl_2_ alone or with 3 mM EDTA ^a^.

	Activity Optimum (°C)	*t*_m_ (DSF) (°C) ^b^		*t*_m_ (CiD) (°C) ^c^			
		2 mM CaCl_2_	2 mM CaCl_2_ + 3 mM EDTA	2 mM CaCl_2_		2 mM CaCl_2_ + 3 mM EDTA	
		** *t* _m_ **	** *t* _m_ **	** *t* _m_ ^N-I^ **	** *t* _m_ ^I-D^ **	** *t* _m_ ^N-I^ **	** *t* _m_ ^I-D^ **
Ika2 ^d^	35	61.97 ± 0.01	50.82 ± 0.01	57.16 ± 1.77	62.10 ± 0.99	47.80 ± 0.23	72.16 ± 0.20
Rho13 ^e^	30	55.21 ± 0.01	53.38 ± 0.01	55.98 ± 0.04	58.32 ± 0.14	N/A ^g^	N/A ^g^
I3C6 ^f^	30	55.40 ± 0.01	52.28 ± 0.01	52.47 ± 0.05	53.85 ± 0.52	55.67 ± 0.25	75.42 ± 0.70

Notes: ^a^ The buffer used was 10 mM MOPS pH 7.5, 50 mM NaCl (CiD), or 50 mM MOPS pH 7.5, 50 mM NaCl (DSF). ^b^ DSF unfolding curves were analyzed as global fits of three technical replicates of 330 nm/350 nm fluorescence and followed a single transition in which the signal of both the native and the denatured state changed linearly with rising temperature. ^c^ CiD was measured as ellipticity of 210–230 nm and showed a three-state unfolding N-I-D, typically with quite narrow intervals of unfolding. The three states were individually fitted as described below with global fits to 21 different wavelengths from 210 to 230 nm (or from nine different wavelengths from 214 to 230 for Ika2_CaCl2_). ^d^ The CiD data for the unfolding of Ika2_CaCl2_ did not converge on a fit with the full 214–230 nm spectrum, but could be fitted with every second wavelength to a satisfying fit using linear baseline correction of the intermediate state. Ika2_EDTA_ was required for baseline correction of all three states. ^e^ Rho13_CaCl2_ was baseline-corrected to a linear change in signal with temperature for the native and denatured states, while Rho13_EDTA_ showed a completely different unfolding curve, likely suggesting denaturation upon stripping of the Ca^2+^. ^f^ I3C6_CaCl2_ required baseline correction of all three states, while I3C6_EDTA_ did not require baseline correction of the native state. ^g^ Not fitted because the data did not show a clear transition.

**Table 3 biomolecules-15-00415-t003:** Average values ± standard error of the mean of multiple structural metrics determined for the continuous 1.2 μs MD trajectories (i.e., three 400 ns–long replicas concatenated into a single trajectory) performed with the three psychrophilic α-amylases (Rho13, Ika2, and I3C6) and their respective homologs (1WZA, 1J0H, and 1HVX).

	Average RMSD (Backbone)/Å	Average RMSF (Backbone)/Å	Average No. H-Bonds	Average *R_g_* (Backbone)/Å	Average SASA/nm^2^
**Rho13**	2.763 ± 0.006	1.02 ± 0.02	413.81 ± 0.09	23.46 ± 0.02	216.56 ± 0.03
**1WZA (homolog)**	2.600 ± 0.004	0.88 ± 0.02	417.01 ± 0.09	22.88 ± 0.02	208.68 ± 0.03
**Ika2**	2.837 ± 0.005	1.295 ± 0.008	1000.6 ± 0.1	33.400 ± 0.002	462.69 ± 0.06
**1J0H (homolog)**	2.123 ± 0.004	1.075 ± 0.008	999.8 ± 0.1	33.467 ± 0.002	458.63 ± 0.06
**I3C6**	1.670 ± 0.003	0.92 ± 0.02	410.79 ± 0.09	24.11 ± 0.02	205.81 ± 0.03
**1HVX (homolog)**	1.073 ± 0.002	0.72 ± 0.01	424.97 ± 0.09	23.73 ± 0.02	186.45 ± 0.02

## Data Availability

The original contributions presented in this study are included in the article and Appendix A. Further inquiries can be directed to the corresponding author.

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
