# Peer review of "Cold-Active Starch-Degrading Enzymes from a Cold and Alkaline Greenland Environment: Role of Ca2+ Ions and Conformational Dynamics in Psychrophilicity"

_biomolecules, 2025, doi:10.3390/biom15030415_

Round 1
Reviewer 1 Report
Comments and Suggestions for Authors
The paper “Cold-active starch-degrading enzymes from a cold and alkaline Greenland environment: role of Ca2+ ions and conformational dynamics in psychrophilicity” by Bendtsen et al. describes three novel secreted amylases from the cold- and alkaline-adapted bacteria. Cold-active enzymes are of high demand for biotechnology because of their activity at ambient temperature and possibility to work with thermo-labile substrates.
The authors describe two novel starch-degrading enzymes from the ikaite columns bacteria and provide detailed characterization of another enzyme obtained from the metagenomic library. Recombinant amylases were expressed in E. coli cells, purified and extensively characterized by different biochemical and biophysical methods. Additionally, their 3D models were obtained and used for molecular dynamics simulations.
The substrate specificity, temperature and pH optimum of the recombinant enzymes were determined. The impact of Ca2+ ion on their activity and stability was demonstrated.
To summarize, described amylases represent interesting examples of cold-active enzymes from extreme environment and in this respect the article has a sufficient impact to the field. The experiments have been carried out properly in most parts. The paper is written in a classic and convincing style.
The authors performed large experimental work to demonstrate the cold-adapted phenotype of the studied enzymes. However, important biochemical characteristics are missing, specifically the catalytic constants. I recommend to determine their Km and Kcat for more detailed characterization. It is known that cold-active enzymes tend to have higher Km values and the catalytic rate (kcat) than their mesophilic homologs.
Some minor points:
Fig. 4 a,b – The axis x name should be T (temperature), not t (time); d – Please specify what does ionic strength (mM) mean. If it is NaCl concentration, the name of the x axis should be modified accordingly.
Author Response
Reviewer 1:
The paper “Cold-active starch-degrading enzymes from a cold and alkaline Greenland environment: role of Ca2+ ions and conformational dynamics in psychrophilicity” by Bendtsen et al. describes three novel secreted amylases from the cold- and alkaline-adapted bacteria. Cold-active enzymes are of high demand for biotechnology because of their activity at ambient temperature and possibility to work with thermo-labile substrates.
The authors describe two novel starch-degrading enzymes from the ikaite columns bacteria and provide detailed characterization of another enzyme obtained from the metagenomic library. Recombinant amylases were expressed in E. coli cells, purified and extensively characterized by different biochemical and biophysical methods. Additionally, their 3D models were obtained and used for molecular dynamics simulations.
The substrate specificity, temperature and pH optimum of the recombinant enzymes were determined. The impact of Ca2+ ion on their activity and stability was demonstrated.
To summarize, described amylases represent interesting examples of cold-active enzymes from extreme environment and in this respect the article has a sufficient impact to the field. The experiments have been carried out properly in most parts. The paper is written in a classic and convincing style.
The authors performed large experimental work to demonstrate the cold-adapted phenotype of the studied enzymes. However, important biochemical characteristics are missing, specifically the catalytic constants. I recommend to determine their Km and Kcat for more detailed characterization. It is known that cold-active enzymes tend to have higher Km values and the catalytic rate (kcat) than their mesophilic homologs.
Response: we appreciate the reviewer’s many helpful comments and agree that it would be insightful to obtain more detailed data on their K(M) and k(cat) values. In fact, the situation is somewhat ambiguous for cryophilic enzymes, with exceptions to the higher KM and kcat values reported e.g. in [1] and [2], to name but a few. We also respectfully note that many publications do not include such data as part of the description of these enzymes, e.g. [3-5] (these references are shown at the end of our response to the reviewer).
We have started planning to collect these data and expect to commence them over the next month or so, but based on our current capabilities, we estimate that it will take 3 months to obtain a complete data set, which is a rather long postponement of the current manuscript. Since this study in any case has Ca2+ dependence and enzyme conformational dynamics as its main focus, we believe that the current study can be read with profit without the inclusion of the aforesaid KM and kcat values. We have therefore taken the liberty of adding the text:
“This allows a direct and simple comparison between the psychrophilic properties of the three enzymes. Detailed mechanistic studies providing KM and kcat values will be reported elsewhere in a separate study.”
straight after the following text:
“In the following, we report on the enzymatic activity of the two amylases as a function of temperature, pH, CaCl2 and NaCl as well as presenting a structural and biophysical analysis in vitro and in silico. We compare them to the psychrophilic amylase I3C6, which was isolated from a metagenomic analysis of ikaite columns and was expressed and purified recombinantly (Fig. 1c). “
Some minor points:
Fig. 4 a,b – The axis x name should be T (temperature), not t (time); d – Please specify what does ionic strength (mM) mean. If it is NaCl concentration, the name of the x axis should be modified accordingly.
Response: thank you, this has been corrected.
REFERENCES:
[1] B. Yin, H. Gu, X. Mo, Y. Xu, B. Yan, Q. Li et al. and C. Jiang, Identification and molecular characterization of a psychrophilic GH1 β-glucosidase from the subtropical soil microorganism Exiguobacterium sp. GXG2, AMB Express 9(1) (2019) 159.
[2] J. Sun, W. Wang, C. Yao, F. Dai, X. Zhu, J. Liu, J. Hao, Overexpression and characterization of a novel cold-adapted and salt-tolerant GH1 β-glucosidase from the marine bacterium Alteromonas sp. L82, Journal of Microbiology 56(9) (2018) 656-664.
[3] X. Wang, G. Kan, X. Ren, G. Yu, C. Shi, Q. Xie et al. and M. Betenbaugh, Molecular Cloning and Characterization of a Novel α-Amylase from Antarctic Sea Ice Bacterium Pseudoalteromonas sp. M175 and Its Primary Application in Detergent, BioMed Research International 2018(1) (2018) 3258383.
[4] J.-W. Zhang, R.-Y. Zeng, Purification and Characterization of a Cold-Adapted α-Amylase Produced by Nocardiopsis sp. 7326 Isolated from Prydz Bay, Antarctic, Marine Biotechnology 10(1) (2008) 75-82.
[5] A.N.M. Ramli, M.A. Azhar, M.S. Shamsir, A. Rabu, A.M.A. Murad, N.M. Mahadi, R. Md. Illias, Sequence and structural investigation of a novel psychrophilic α-amylase from Glaciozyma antarctica PI12 for cold-adaptation analysis, Journal of Molecular Modeling 19(8) (2013) 3369-3383.
Reviewer 2 Report
Comments and Suggestions for Authors
The review concerns the manuscript “Cold-active starch-degrading enzymes from a cold and alkaline Greenland environment: role of Ca2+ ions and conformational dynamics in psychrophilicity” written by Bendtsen, M. K., et al., and submitted to Biomolecules.
The paper generally touches on an essential topic of psychrozymes and their properties, which is crucial in understanding the adaptations of living organisms to their environments. The authors describe three novel secreted amylases Rho13, Ika2 and I3C6, all 20 from bacteria growing in the cold and alkaline ikaite columns in Greenland. The research employs both wet lab work and molecular dynamics simulations. The manuscript is well written and properly illustrated. I believe that the topic may be definitely attractive to the journal’s readership as it sheds light on the many ways in which psychrophilic enzymes adapt to increased catalysis at lower temperatures. I do not have any major issues regarding the methodology or results, therefore, I recommend accepting the manuscript.
My minor comments are:
- Ca2+ in the title should probably be Ca2+.
- Perhaps Figure S1 could be transferred to the main manuscript, as it gives the reader the basic information about the folds of the enzymes, at which the work is based on.
Author Response
Reviewer 2:
The review concerns the manuscript “Cold-active starch-degrading enzymes from a cold and alkaline Greenland environment: role of Ca2+ ions and conformational dynamics in psychrophilicity” written by Bendtsen, M. K., et al., and submitted to Biomolecules.
The paper generally touches on an essential topic of psychrozymes and their properties, which is crucial in understanding the adaptations of living organisms to their environments. The authors describe three novel secreted amylases Rho13, Ika2 and I3C6, all 20 from bacteria growing in the cold and alkaline ikaite columns in Greenland. The research employs both wet lab work and molecular dynamics simulations. The manuscript is well written and properly illustrated. I believe that the topic may be definitely attractive to the journal’s readership as it sheds light on the many ways in which psychrophilic enzymes adapt to increased catalysis at lower temperatures. I do not have any major issues regarding the methodology or results, therefore, I recommend accepting the manuscript.
My minor comments are:
- Ca2+ in the title should probably be Ca2+.
- Perhaps Figure S1 could be transferred to the main manuscript, as it gives the reader the basic information about the folds of the enzymes, at which the work is based on.
Response: Thank you for these encouraging responses. We have addressed both points as requested.